# Long-Term Observation of Atmospheric Speciated Mercury during 2007–2018 at Cape Hedo, Okinawa, Japan

**Kohji Marumoto [1,*], Noriyuki Suzuki [2], Yasuyuki Shibata [3], Akinori Takeuchi [3], Akinori Takami [4], Norio Fukuzaki [5], Kazuaki Kawamoto [6], Akira Mizohata [7], Shungo Kato [8], Takashi Yamamoto [9], Jingyang Chen [9], Tatsuya Hattori [10], Hiromitsu Nagasaka [10] and Mitsugu Saito [11]**

[1]   Department of Environment and Public Health, National Institute for Minamata Disease (NIMD), 4058-18, Hama, Minamata-shi, Kumamoto 867-0008, Japan

[2]   Center for Health and Environmental Risk Research, National Institute for Environmental Studies (NIES), 16-2, Onogawa, Tsukuba-shi, Ibaraki 305-8506, Japan

[3]   Center for Environmental Measurement and Analysis, National Institute for Environmental Studies (NIES), 16-2, Onogawa, Tsukuba-shi, Ibaraki 305-8506, Japan

[4]   Center for Regional Environmental Research, National Institute for Environmental Studies (NIES), 16-2, Onogawa, Tsukuba-shi, Ibaraki 305-8506, Japan

[5]   Department of Environmental Science, Niigata Institute of Technology, 1719, Fujihashi, Kashiwazaki-shi, Niigata 945-1195, Japan

[6]   Graduate School of Fisheries and Environmental Sciences, Nagasaki University, 1-14, Bunkyo-machi, Nagasaki-shi, Nagasaki 852-8521, Japan

[7]   Professor Emeritus, Osaka Prefecture University, Gakuen-cho, Naka-ku, Sakai-shi, Osaka 599-8570, Japan

[8]   Department of Applied Chemistry for Environment, Tokyo Metropolitan University, 1-1, Minami-Osawa, Hachioji-shi, Tokyo 192-0397, Japan

[9]   Institute for Environmental Informatics, IDEA Consultants, Inc. 2-2-2, Hayabuchi, Tsuzuki-ku, Yokohama-shi, Kanagawa 224-0025, Japan

[10]  Institute for Environmental Ecology, IDEA Consultants, Inc. 1334-5, Riemon, Yaizu-shi, Shizuoka 421-0212, Japan

[11]  Office of Mercury Management, Environmental Health Department, Ministry of the Environment Government of Japan, 1-2-2, Kasumigaseki, Chiyoda-ku, Tokyo 100-8975, Japan

*   Correspondence: marumoto@nimd.go.jp; Tel.: +81-966-63-3111; Fax: +81-966-63-7822

**Abstract:** The concentrations of atmospheric gaseous elemental mercury (GEM), gaseous oxidized mercury (GOM), and particle-bound mercury (particles with diameter smaller than 2.5 μm; $PBM_{2.5}$) were continuously observed for a period of over 10 years at Cape Hedo, located on the north edge of Okinawa Island on the border of the East China Sea and the Pacific Ocean. Regional or global scale mercury (Hg) pollution affects their concentrations because no local stationary emission sources of Hg exist near the observation site. Their concentrations were lower than those at urban and suburban cities, as well as remote sites in East Asia, but were slightly higher than the background concentrations in the Northern Hemisphere. The GEM concentrations exhibited no diurnal variations and only weak seasonal variations, whereby concentrations were lower in the summer (June–August). An annual decreasing trend for GEM concentrations was observed between 2008 and 2018 at a rate of $-0.0382 \pm 0.0065$ ng m$^{-3}$ year$^{-1}$ ($-2.1\% \pm 0.36\%$ year$^{-1}$) that was the same as those in Europe and North America. Seasonal trend analysis based on daily median data at Cape Hedo showed significantly decreasing trends for all months. However, weaker decreasing trends were observed during the cold season from January to May, when air masses are easily transported from the Asian continent by westerlies and northwestern monsoons. Some GEM, GOM, and $PBM_{2.5}$ pollution events were observed more frequently during the cold season. Back trajectory analysis showed that almost all these events

occurred due to the substances transported from the Asian continent. These facts suggested that the decreasing trend observed at Cape Hedo was influenced by the global decreasing GEM trend, but the rates during the cold season were restrained by regional Asian outflows. On the other hand, GOM concentrations were moderately controlled by photochemical production in summer. Moreover, both GOM and PBM$_{2.5}$ concentrations largely varied during the cold season due to the influence of regional transport rather than the trend of atmospheric Hg on a global scale.

**Keywords:** atmospheric Hg; speciation; long-term observation; annual trend; Japan; East Asia

## 1. Introduction

Toxic trace metals, including mercury (Hg), cadmium (Cd), and lead (Pb), that are emitted from anthropogenic and natural sources, exist in the natural environment for long periods of time without degradation. A large part of atmospheric Hg is present as gaseous elemental mercury (GEM), but gaseous oxidized mercury (GOM) and particle-bound mercury (PBM) also exist in the atmosphere [1–3]. These forms of Hg are emitted from various sources, including power plants, depending on the flue cleaning systems [4]. Gaseous elemental mercury has a long atmospheric residence time because it does not readily dissolve in atmospheric water, including rain and snow, and has a slow oxidation and possible GOM reduction [4]. Therefore, GEM is readily transported in the atmosphere and deposited to regions farther from the emission sources compared to GOM and PBM. On the other hand, GOM and PBM are easily deposited via wet and dry deposition processes [1,5]. Gaseous oxidized mercury is mainly produced by photochemical reactions between GEM and oxidants, such as $O_3$, and OH and Br radicals in the atmosphere [6,7]. In addition, the adsorption of GEM and GOM onto the surfaces of atmospheric suspended particles also leads to PBM formation [8–10]. Then, they concentrate within organisms that inhabit exposed lands and oceans because of specific behaviors, such as gas exchange through the stoma of vegetation, in situ methylation in various environmental mediums, and bioaccumulation of methyl mercury in the aquatic nutritional chain.

Currently, the three types of Hg species in the atmosphere can be measured using an automated Tekran instrument [11] and some monitoring networks, including CAMNet (Canadian Atmospheric Mercury Network), AMNet (Atmospheric Mercury Network), and GMOS (Global Mercury Observation System). These systems have been continually used for the atmospheric Hg monitoring mainly in North America and European regions [1,12–15]. In these networks, there are background sites that store monitoring data for more than a few decades in order to evaluate the global Hg cycle, as well as the long-term trends of Hg levels in the atmosphere. For example, total gaseous mercury (TGM, is similar to GEM) levels have been measured at Mace Head, Ireland since September 1995; the monitoring data collected here revealed that there has been a decrease in TGM concentrations globally at a rate of 1.6–2.0% per year since 1996 [16,17], although atmospheric Hg concentrations increased in the late 1970s and 1980s [18]. In addition, a significant decrease in annual GEM concentrations was observed at Alert, Canada, at a rate of 0.6% year$^{-1}$ from 1995 to 2007 [19]. At Cape Point in South Africa, TGM was measured by a manual sampling method using a gold amalgamation trap for about 10 years from September 1995, and GEM was measured using an automated Tekran 2537B monitor since March 2007 [20,21]. The data thus obtained on TGM (GEM) concentrations at this station revealed a decreasing trend of concentrations from 1995 to 2005 and an increasing trend since 2007; seasonal variation in these concentrations indicated the influence of biomass burning in South America and southern Africa [21]. On the other hand, GEM concentrations at Zeppelin station, Ny-Alesund, Svalbard showed no distinct annual trend during the period from 2000 to 2009 [22]. These flat and/or decreasing trends of atmospheric Hg concentrations are inconsistent with the slightly increasing trend of anthropogenic emissions worldwide [23]. This emission trend is, however, consistent with the increasing trend of atmospheric Hg concentrations at Cape Point since 2007. One of the reasons for the decreasing

atmospheric Hg concentrations in the world is proposed to be its reduced re-emission from the legacy of historically deposited mercury in oceans and soil reservoirs [24]. In addition, Soerensen et al. [25] suggested that the decline in atmospheric Hg concentrations, at least in the Northern hemisphere, was caused by the decreasing Hg concentrations in the subsurface of the North Atlantic Ocean. Moreover, an improved global Hg emission inventory for the period from 1990 to 2010 revealed that the observed decline in atmospheric Hg concentrations can be explained by the decrease in the anthropogenic $Hg^0$ emissions, with much larger reductions in Europe and North America [26]; this is considering the decline in atmospheric release of Hg mainly from commercial products and the prevalence and improvement of emission controls on coal fired utilities. Meanwhile, the global anthropogenic Hg emissions increased from the year 2000 to 2010 with 68% of the emissions being released from Asia and Oceania in 2010 [26]. In addition, the anthropogenic Hg emissions in Asia have constantly increased from 1995 until now; Asia was responsible for more than 50% of the global anthropogenic emissions in 2010 [27]. Thus, long-term monitoring activities are necessary for evaluation of the continuous influence of Asian emissions and to gauge the magnitude of the outflow of atmospheric Hg from Asia. However, such data is scarce from many Asian regions. The Lulin Atmospheric Background Station (LABS) in Taiwan is the only background site in Asia that has been continuously measuring the concentrations of GEM, GOM, and PBM in the atmosphere for more than 10 years [28].

In this study, we conducted measurements of atmospheric Hg at the station in Cape Hedo, Okinawa since October 2007 and recorded the data for about 10 years. Cape Hedo is located on the north edge of Okinawa Island on the border of the East China Sea and the Pacific Ocean (Figure 1). The northwestern monsoon prevails at this site during the cold season from late fall to early spring, implying that the air pollutants emitted from the Asian continent are easily transported here. Many researchers have reported the regional air pollution events of $O_3$ [29], carbon monoxide [30], black carbon [31–33], polycyclic aromatic hydrocarbons (PAHs) [34], $PM_{2.5}$ [35,36], metallic elements [36,37], and GEM [5,38] etc. at this site. We investigated the long-term trends of atmospheric speciated Hg at Cape Hedo as it is one of the background sites in Asia; we used our data from this site to reveal the relationship between Hg emissions and long-range transport on a regional scale in East Asia.

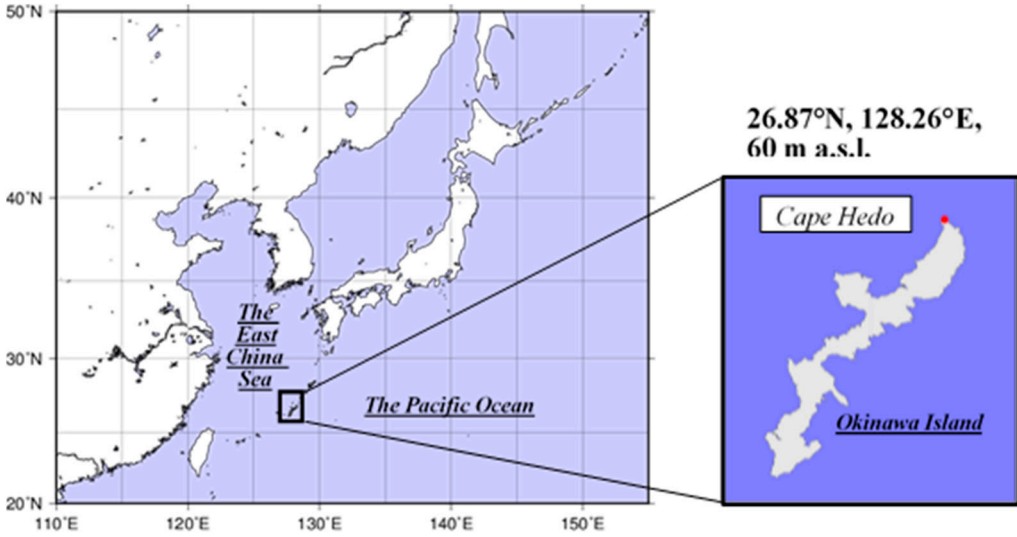

**Figure 1.** A map of the observation site.

## 2. Materials and Methods

### 2.1. Site Information

A map showing the location of the sampling site in Cape Hedo, Okinawa is shown in Figure 1. The Cape Hedo Atmosphere and Aerosol Monitoring Station (CHAAMS; 26.87° N, 128.26° E, 60 m

above sea level) is a national monitoring site established and operated by the National Institute for Environmental Studies (NIES) in order to study transboundary air pollution. The site is surrounded by the sea on three sides. Within 25 km radius of this site, there are no local stationary emission sources, such as coal-fired electric power plants, municipal waste incinerators, or factories. Air masses come to this site more frequently from mainland China and the Korean Peninsula, especially during cold seasons, i.e., from December to May. In addition, air masses from the Philippines and/or the Pacific Ocean arrive at the site, especially during the warm season (June–November). Therefore, it is a suitable location for investigating the distribution and transportation properties of hazardous metals in the East Asia region. Meteorological parameters, including wind direction and speed, air temperature, relative humidity, air pressure, and rainfall were monitored using a WXT520 (Vaisala Co. Ltd., Helsinki, Finland) at this site. Carbon monoxide (CO) and $O_3$ were also measured by Thermo Environmental Instruments Model 48C and 49i, respectively [30].

## 2.2. Atmospheric Speciated Hg Measurements

Continuous measurements of atmospheric GEM concentrations were started in October 2007 using an automated ambient air mercury monitor manufactured by Tekran Inc. (Model 2537B or 2537X, Toronto, Canada). In addition, from October 2009 to December 2018, the concentrations of GOM and particle-bound mercury of particles with diameter smaller than 2.5 μm ($PBM_{2.5}$) were also measured by using the Tekran speciation units (Model 1130, and Model 1135). However, the GEM observation data was not recorded from mid-May 2008 to July 2008 due to some troubles in the monitor. Ambient air containing GEM, GOM, and $PBM_{2.5}$ was drawn at a flow rate of 10 L min$^{-1}$ through an air inlet with an impactor to remove coarse particles larger than 2.5 μm. Firstly, GOM was adsorbed on the interior surface of a KCl-coated annular denuder and any $PBM_{2.5}$ were captured by a quartz fiber filter present downstream of the denuder. During the sampling of GOM and $PBM_{2.5}$, GEM was measured every 5 min using two gold cartridges that alternately collected and thermally desorbed Hg, which was subsequently measured by cold-vapor atomic fluorescence spectrometry (CVAFS). The speciation system was operated on a 3-h cycle that comprised steps of GEM measurement and pre-concentration of GOM and $PBM_{2.5}$ for 2 h, followed by GOM and $PBM_{2.5}$ analysis for an hour. The CVAF spectrometer was automatically calibrated once a day using an internal permeation tube. The method detection limit (MDL) for GEM is less than 0.1 ng m$^{-3}$ [39]; all measured values were more than 10 times higher than the MDL. The MDLs for GOM and $PBM_{2.5}$ were approximately 1 pg m$^{-3}$ as determined based on their blank values recorded from the third reading during the zero-air flush. The MDLs fluctuated depending on system stability. The measured concentrations of GOM and $PBM_{2.5}$ were, in some instances, less than or equal to their MDLs, accounting for over 70% of the GOM and about 30% of the $PBM_{2.5}$ of the total measurements. Therefore, their mean and median concentrations were calculated by substituting the values that were less than the MDLs with half the values of MDLs, i.e., 0.5 pg m$^{-3}$. Temporal trends in GEM concentrations were calculated using a least-squares method and the rates were obtained from the slope of the regression line.

## 2.3. QA/QC Program for Speciated Hg

To implement quality assurance and quality control on the measurement of atmospheric Hg via the Tekran Hg speciation system, we prepared our own standard operating procedure (SOP) based on US Environmental Protection Agency's (EPA's SOP). Briefly, to maintain stable operation under conditions of high temperature and humidity in Cape Hedo, Okinawa, a high-power air dryer was installed to prevent dropwise condensation in the supply route of zero air, and the flappers of the speciation units were sealed using an aluminum tape to eliminate highly humid incoming air. In addition, operation was shut down to prevent incoming seawater and/or sea salts from entering the system just before typhoons approached; it was re-started as quickly and safely as possible when they passed away. The detector was automatically calibrated once a day using an internal permeation tube for GEM. In addition, to checking the auto-calibration system, external manual injections using a

saturated Hg standard gas (MB-1, Nippon Instruments Co. Ltd., Takatsuki, Japan) were performed twice in a year. Moreover, the measured GEM values by the Tekran system were compared with those obtained by an official manual method by the Ministry of the Environment, Japan (MOEJ) [40] with some modification to ensure the quality of data derived from the Tekran monitoring system. In our manual method, a soda lime column was installed upstream of a gold amalgamation trap to prevent moisture during air sampling. Figure 2a shows a scatter plot of both the data, depicting their good correlation. In addition, to compare the atmospheric concentration of GOM and $PBM_{2.5}$ obtained by Tekran monitoring system and the offline manual method [11], a scatterplot is shown in Figure 2b. In the manual method, a thermal desorption-gold amalgamation-CVAFS (RA-FG+, Nippon Instruments Co. Ltd.) and a thermal desorption-gold amalgamation-cold-vapor atomic absorption spectrometry (CVAAS) CVAAS (MA-2, Nippon Instruments Co. Ltd., Mumba, India) were used for GOM and $PBM_{2.5}$ analyses, respectively. The correlations between the two datasets were in line with each other.

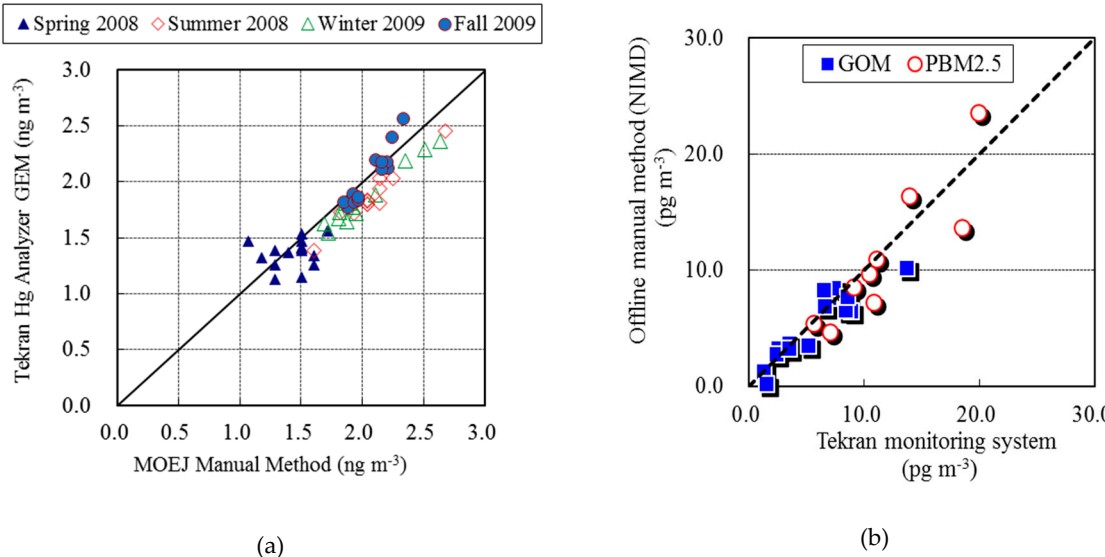

(a)                                (b)

**Figure 2.** Scatterplots of the (**a**) gaseous elemental mercury (GEM) concentrations recorded using the Tekran monitoring system and the total gaseous mercury (TGM) obtained using the method outlined in the official Ministry of the Environment, Japan (MOEJ) manual and (**b**) gaseous oxidized mercury (GOM) and particle-bound mercury for particles with diameter smaller than 2.5 μm ($PBM_{2.5}$) concentrations recorded using the Tekran monitoring system and the offline method.

*2.4. Back Trajectory Analysis*

To identify the transport patterns of air masses arriving at the observation site, the hybrid single-particle Lagrangian integrated trajectory (HYSPLIT) model [41] was used to obtain five-day backward trajectories based on meteorological data available at the Global Data Assimilation System (GDAS). Trajectories were made out at the starting height of 250 m above the observation site at Cape Hedo, beginning at the time when the maximum concentrations of GEM, GOM, and $PBM_{2.5}$ were observed during each pollution event defined below (Section 3.2). In addition, the calculation was carried out at 1 h intervals.

## 3. Results and Discussion

*3.1. Concentrations of GEM, GOM, and $PBM_{2.5}$*

The mean concentrations (±1σ) of GEM, GOM, and $PBM_{2.5}$ during the entire observation period from the year 2007 to 2018 were $1.81 \pm 0.43$ ng m$^{-3}$ (hourly mean values for 5 min intervals, N = 53,840), $1.7 \pm 2.9$ pg m$^{-3}$ (N = 22,071) and $2.6 \pm 3.6$ pg m$^{-3}$ (N = 22,058), respectively (Table 1). The mean GEM concentrations were lower than those reported for urban, suburban, and remote sites in the East

Asian region [15,42–45], but were slightly higher than the background concentrations recorded in the Northern hemisphere (1.3–1.6 ng m$^{-3}$) and in the Southern hemisphere (around 1.0 ng m$^{-3}$) [14,46]. Thus, it was inferred that the concentrations of Hg at Cape Hedo were elevated due to regional pollution because there are no local Hg stationary emission sources around this site. On the other hand, the mean concentrations of GOM and PBM$_{2.5}$ at the site were lower than those in Fukuoka in Japan [44], the Bohai Sea, and Yellow Sea areas [47]; they were less than one-tenth of concentrations observed in Asian urban cities, such as Seoul in Korea [48], and at remote sites in China [49]. The mean fractions of GOM and PBM$_{2.5}$ in the total atmospheric Hg (GEM + GOM + PBM$_{2.5}$) at Cape Hedo were both less than 1.0%. The dominant species of Hg in the atmosphere was GEM at this site as observed for other ground-based observation sites.

**Table 1.** Mean ± standard deviation (SD), median, minimum and maximum concentrations of GEM, GOM, and PBM$_{2.5}$ at Cape Hedo in the entire period from October 2007 to December 2018.

|  | GEM (ng m$^{-3}$) | GOM (pg m$^{-3}$) | PBM$_{2.5}$ (pg m$^{-3}$) |
|---|---|---|---|
| N | 53,840 | 22,071 | 22,058 |
| Mean ± S.D. | 1.81 ± 0.43 | 1.7 ± 2.9 | 2.6 ± 3.6 |
| Minimum | 0.8 | 0.5 | 0.5 |
| 25 percentile | 1.5 | 0.5 | 0.5 |
| Median | 1.7 | 0.5 | 1.4 |
| 75 percentile | 2.0 | 1.8 | 3.1 |
| Maximum | 7.3 | 58 | 71 |

*3.2. Seasonal Variations in the Concentrations of GEM, GOM, and PBM$_{2.5}$*

The monthly mean and median values of GEM, GOM, and PBM$_{2.5}$ during the entire observation period are shown in Figure 3. The GEM concentrations were slightly higher in winter (December–February), spring (March–May), and fall (September–November) as compared to those in summer (June–August). The PBM$_{2.5}$ concentrations were also higher during the cold season than during the warm season, whereas the GOM concentrations were higher in summer than in other seasons.

Figure 4 shows the frequency of the pollution events of GEM, GOM, and PBM$_{2.5}$ that were defined as events, where their concentrations exceeded the mean + 3× standard deviation value. Based on this definition, 176, 172, and 144 events of GEM, GOM, and PBM$_{2.5}$ pollution were identified during the entire observation period, respectively. Figures 5–7 present the results of the back-trajectory analysis for each season in which the GEM, GOM, and PBM$_{2.5}$ pollution events had occurred. Furthermore, 60.5% of the total GOM pollution events occurred during the warm season. It is well known that GOM is produced by the photochemical oxidation of GEM with O$_3$, and BrO, OH, Br, and Cl radicals [50]. In the marine boundary layer, the Br and BrO radicals that originate from sea salt aerosols act as the dominant oxidants for GEM [50,51]. As shown in Figure 6, the air mass came from oceanic regions when the GOM pollution events were observed in summer. Figure 8 also shows the frequency of typhoon passage through the Okinawa region [52]. A lot of typhoons come to this region in summer and early fall; the strong typhoon winds disturb the sea surface, which leads to a more abundant spread of sea salt aerosols into the atmosphere and above land surfaces. At the time that the typhoons approached during our study, Hg measurements were stopped for system preservation and no data was recorded. However, after the typhoons passed, sea salts pervaded the atmosphere and land surfaces around the site. Once they are dried by the strong sunlight, large amounts of oxidants, such as Br and Cl radicals are thought to be released into the atmosphere, resulting in GEM conversion to GOM. Therefore, the seasonal variations in GOM concentrations were consistent with the meteorological conditions at Cape Hedo.

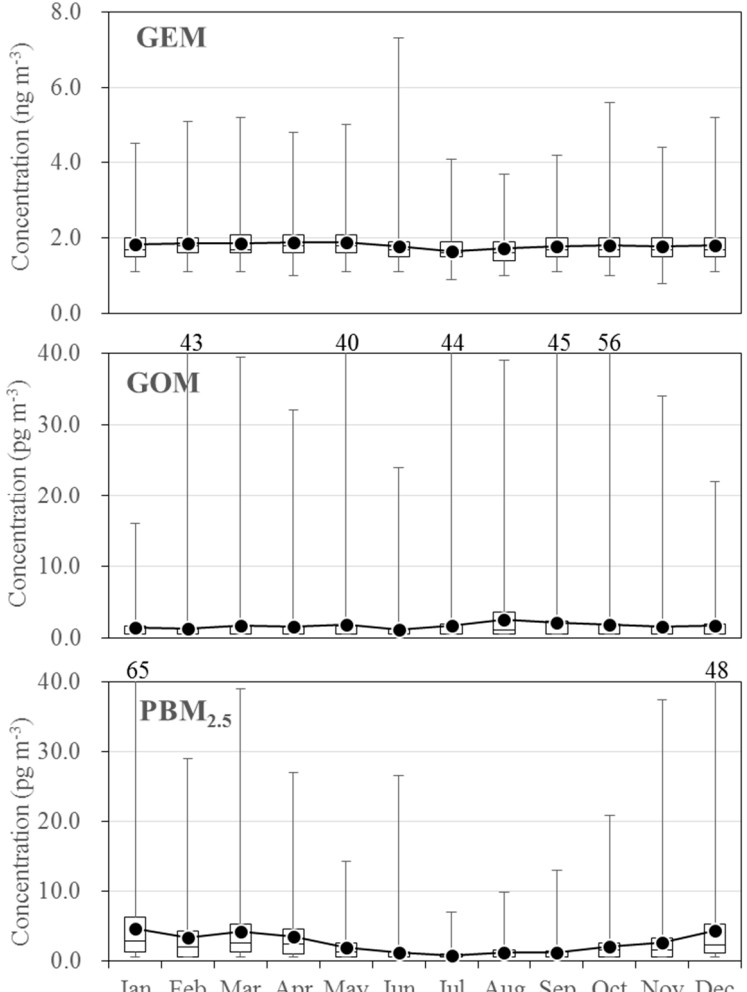

**Figure 3.** Box-whisker plots of the concentration distributions of GEM, GOM, and PBM$_{2.5}$ at Cape Hedo. The middle line in the box and the black circles shows median and mean concentrations, respectively. The box represents the range between the 25th and 75th percentiles and the whisker above and below indicates maximum and minimum concentrations, respectively. The numbers represent the maximum concentrations of GOM and PBM$_{2.5}$ in each month.

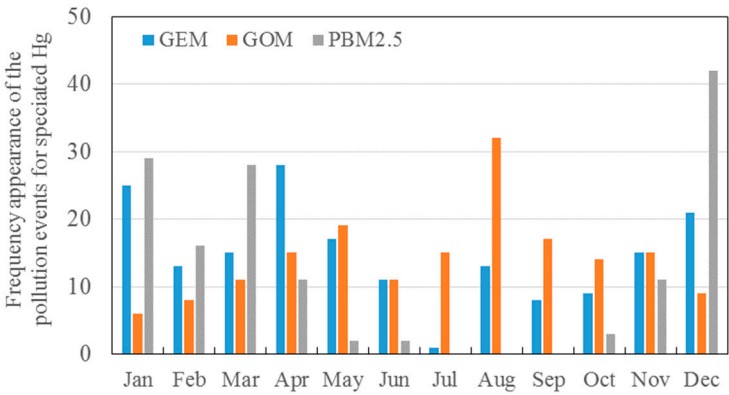

**Figure 4.** Seasonal variations in the frequency of the pollution events of GEM, GOM, and PBM$_{2.5}$. The pollution events are defined as their values above the mean value plus 3× SD.

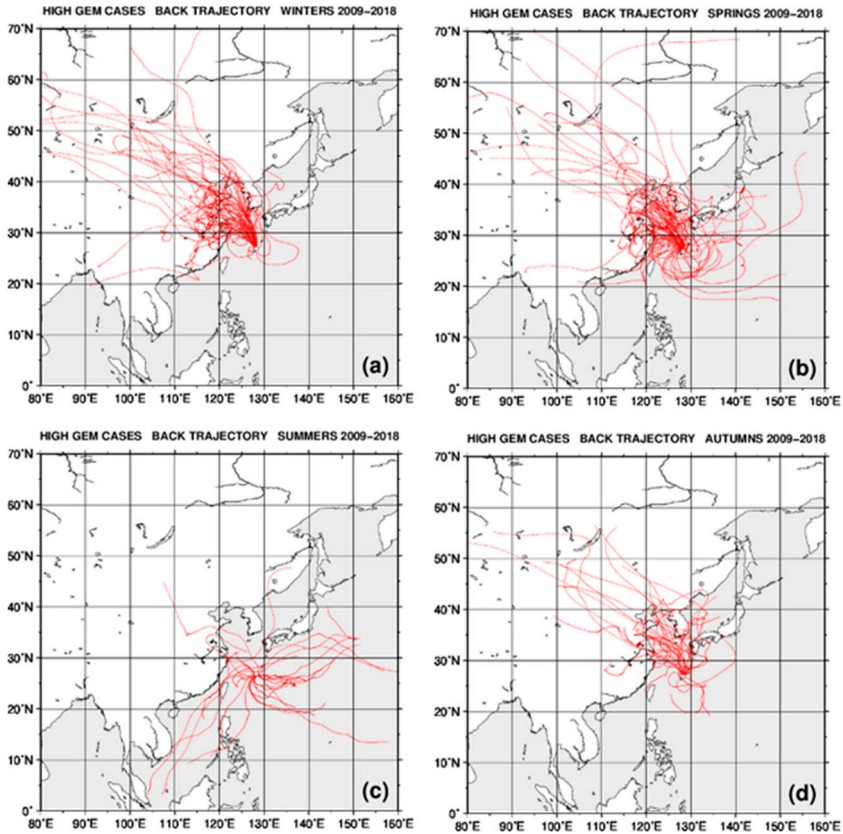

**Figure 5.** Seasonal changes in the pathway of the air mass arriving at Cape Hedo when GEM pollution events were observed in (**a**) winter, (**b**) spring, (**c**) summer, and (**d**) fall.

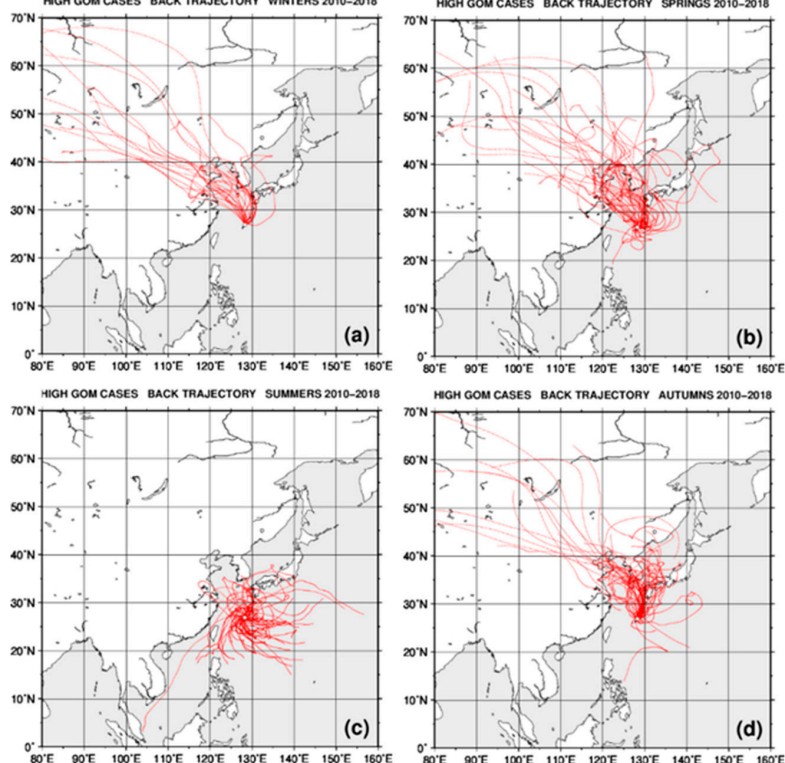

**Figure 6.** Seasonal changes in the pathway of the air mass arriving at Cape Hedo when GOM pollution events were observed in (**a**) winter, (**b**) spring, (**c**) summer, and (**d**) fall.

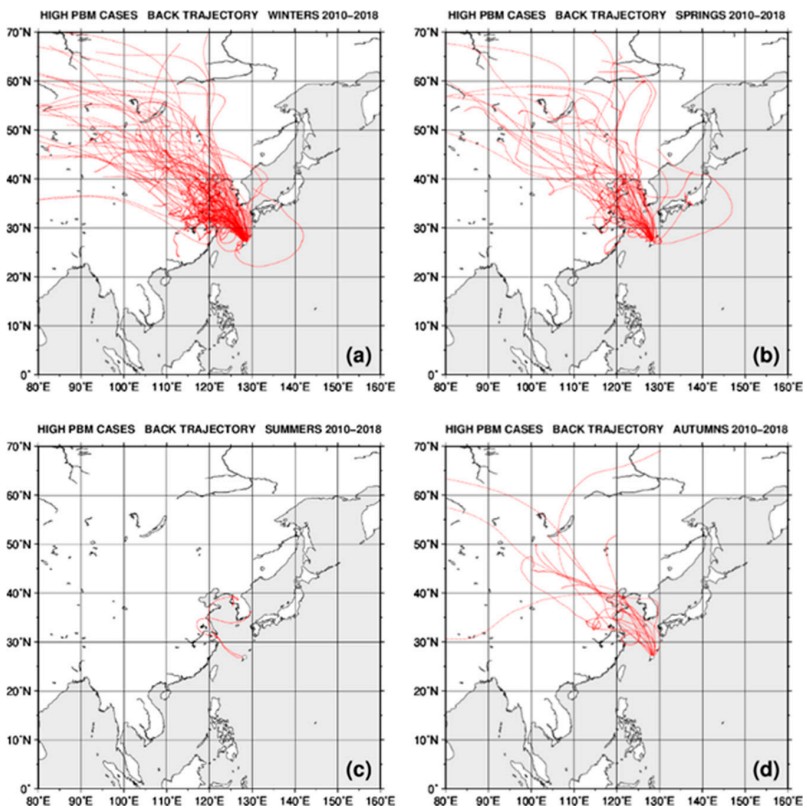

**Figure 7.** Seasonal changes in the pathway of the air mass arriving at Cape Hedo when PBM$_{2.5}$ pollution events were observed in (**a**) winter, (**b**) spring, (**c**) summer, and (**d**) fall.

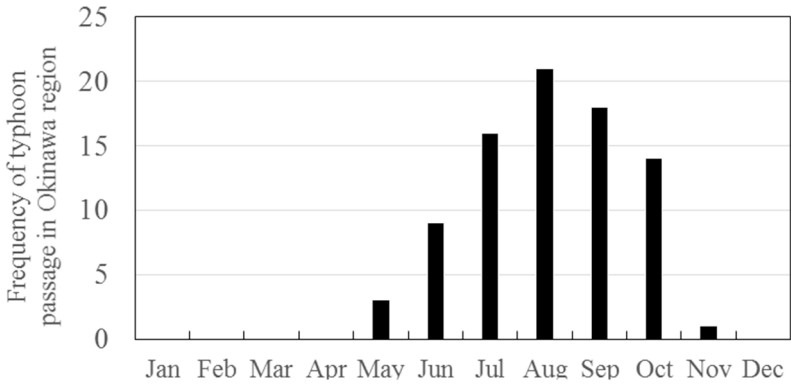

**Figure 8.** Frequency of the typhoon passage through the Okinawa region.

On the other hand, 67.6% of all GEM pollution events (119 events) and 88.9% of the PBM$_{2.5}$ events (128 events) occurred during the cold season. The GOM pollution events were also observed in this season, although they were less frequent than during the warm season. During the cold season (winter and spring) and fall, air masses were transported from the Asian continent more frequently than in summer. The concentrations of GEM, GOM, PBM$_{2.5}$, CO, O$_3$, and other meteorological parameters from December 2013 to January 2014 are shown in Figure 9 as a typical time-series depicting examples of variations in these parameters.

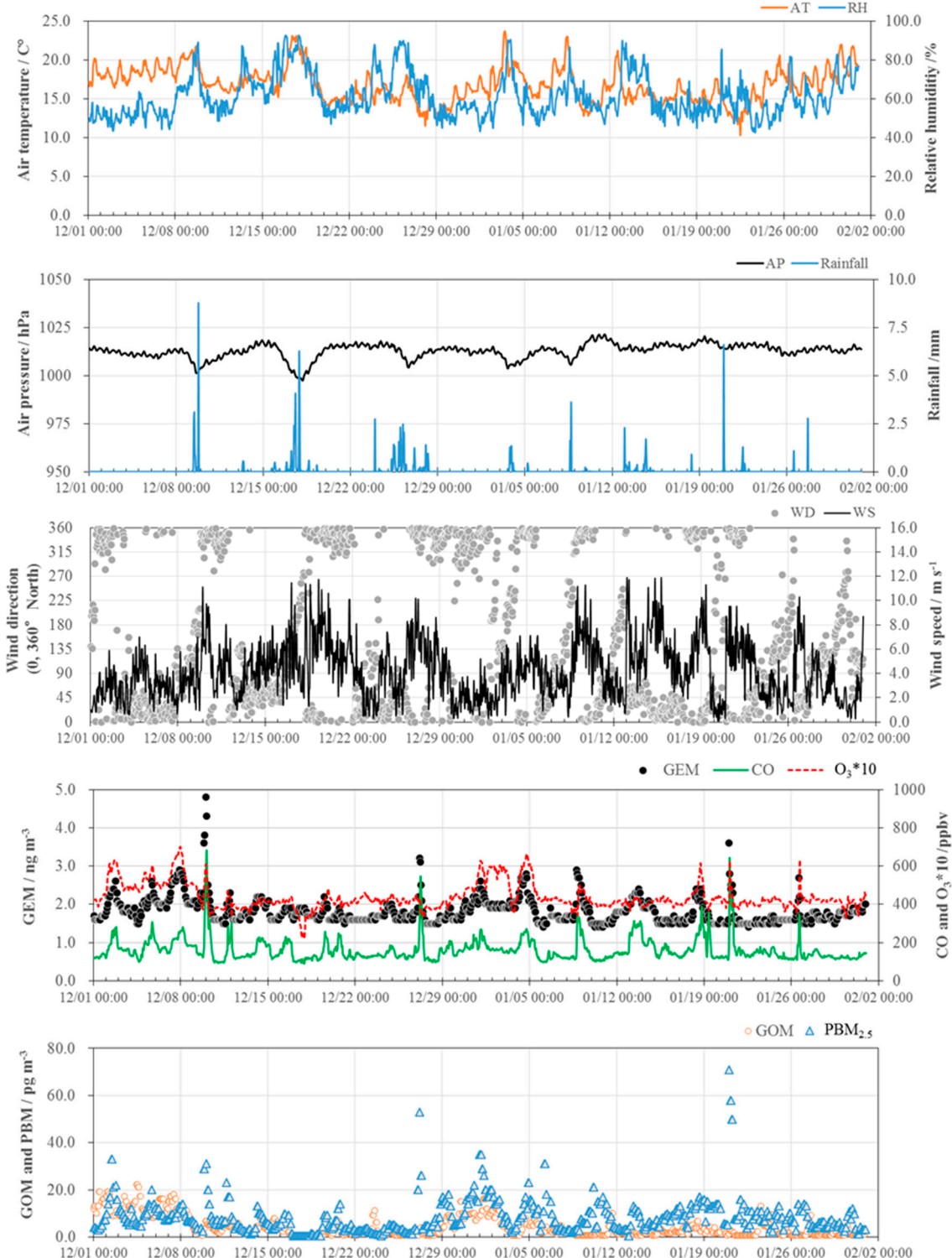

**Figure 9.** Typical examples of time series observed in the concentrations of GEM, GOM, PBM$_{2.5}$, CO, and O$_3$ and meteorological parameters, including air temperature (AT), relative humidity (RH), air pressure (AP), rainfall, wind direction (WD) and wind speed (WS) during the period from December 2013 to January 2014.

The concentrations of GEM, GOM, PBM$_{2.5}$, CO, and O$_3$ had similar variations. They showed remarkably synchronous increases several times, when the air pressure decreased, along with reduced air temperature and relative humidity; thereafter, the wind direction gradually changed from south

to north and the wind speed increased. These meteorological changes indicated that the cold front accompanying an extratropical cyclone—that carried cold and dry air from the Asian continent—had passed over the site. The back trajectories also demonstrated that air masses had come from the Asian continent during almost all such events. In most of these pollution events, the concentrations of GEM, GOM, and $PBM_{2.5}$ had a significant positive correlation with CO and $O_3$, indicating regional Hg transport from the Asian continent [5,28,38,44]. The correlation between GOM and $O_3$ during the pollution events also indicated that aged air masses with photochemical formation of $O_3$ and possibly of GOM was transported from the continent. The pollution events associated with the passage of the cold front or cyclones have been observed at other sites in Japan, such as the GEM event at Fukuoka and the TGM event at Minamata in the Kyushu Islands [44], and events for other air pollutants, such as $PM_{2.5}$ [53,54].

### 3.3. Long-Term Trends of Atmospheric GEM, GOM, and $PBM_{2.5}$ Concentrations

Figure 10 shows the time series of the monthly median GEM concentrations between October 2007 and December 2018. A distinct decreasing trend of GEM concentrations at Cape Hedo was recorded during the observation period. The rate of decrease in the median values was $-2.1\% \pm 0.36\%$ year$^{-1}$. This value was almost comparable to concentrations reported in North America ($-1.5\% \pm 0.15\%$ year$^{-1}$) and Western Europe ($-2.0\% \pm 0.46\%$ year$^{-1}$) for the time period between 1990 and 2010 [26], although an increasing trend of GEM concentrations was observed at Cape Point in South Africa between 2007 and 2015 [21]. Soerensen et al. [25] suggested that the decline in atmospheric Hg concentrations in the Northern hemisphere was caused by the decrease in oceanic evasion from the North Atlantic, that is accompanied by a decrease in oceanic Hg concentrations in the subsurface layer. Using a coupled global atmosphere–ocean model, they also found out that the decline in oceanic Hg concentrations in the North Atlantic affects the global Hg budget. On the other hand, using an improved global Hg emission inventory, Zhang et al. [26] found that the decrease in global anthropogenic Hg emissions affects the decline in atmospheric Hg. The Japan Coast Guard investigated the total Hg concentrations in surface water in the East China Sea in recent years. Figure 11 shows the annual trend of Hg concentrations [55]. The oceanic Hg concentrations decreased in recent years at a rate of $-0.032$ ng L$^{-1}$ year$^{-1}$ ($-15.7\%$ year$^{-1}$) in the East China Sea. This rate was steeper as compared to the atmospheric GEM decline, probably because of the shorter observation period and small sample size (three sites in each year). In addition, the analytical method to measure Hg used in this survey was slightly different from the one recommended by international standard methods, such as the EPA method 1631 [56]. However, this indicates that a similar phenomenon can occur not only in the North Atlantic Ocean but also in the East China Sea.

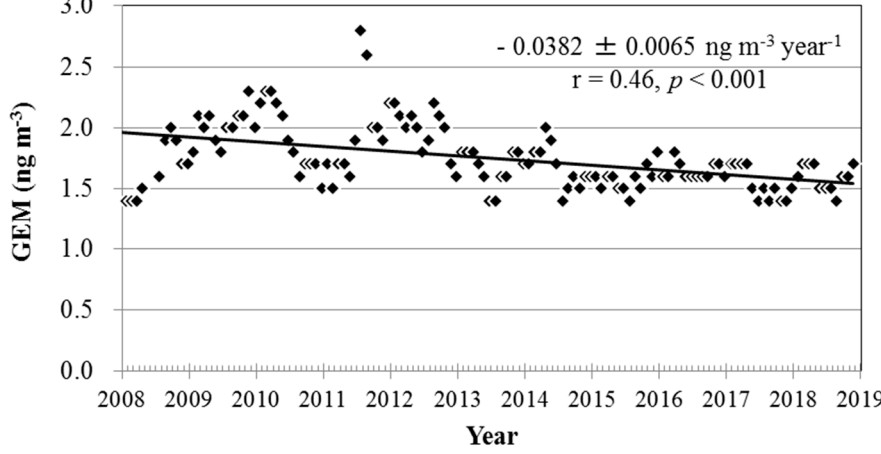

**Figure 10.** Long-term variations in the monthly mean GEM concentrations. The slope of the least-square regression represents its annual trend.

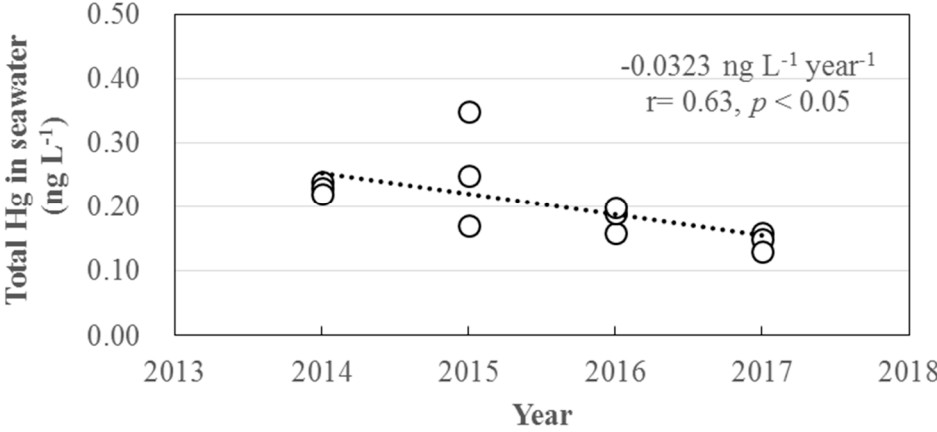

**Figure 11.** The variations in the total Hg concentrations in surface seawater of the East China Sea between 2014 and 2017. The slope of the least-square regression represents its annual trend.

The results of the seasonal trend analysis for every month based on daily median data on GEM concentrations at Cape Hedo are presented in Figure 12. A negative slope indicates a decreasing trend during each month from 2008 to 2018. At this site, a decreasing trend, ranging from $-0.0695 \pm 0.0065$ ng m$^{-3}$ year$^{-1}$ (September) to $-0.0115 \pm 0.0069$ ng m$^{-3}$ year$^{-1}$ (April) was observed for all months. In addition, the months in the cold season had weaker decreasing trends than those in the warm season. As mentioned before, during the cold season, air masses that carry some air pollutants from the Asian continent are transported to our monitoring site by westerlies and northwestern monsoons. In the East Asian region, China is one of the major contributors of atmospheric Hg emissions [57], although the estimated emission amounts [58] and emission outflow [59] have been reported to have slightly decreased due to emission controls in recent years. In addition, the Hg emissions in East Asia slightly increased in the period between 2010 (901.4 Mg) and 2015 (1012.3 Mg), although its growth rate has slowed in recent years [60]. Higher GEM concentrations at Cape Hedo are still observed more frequently during the cold season due to the Asian outflows as previously mentioned. Therefore, it is suggested that the annual GEM concentrations at Cape Hedo decreased due to the global decreasing trends, especially those recorded in Europe and North America; the magnitude of this decrease was restrained by the outflow from the Asian continent.

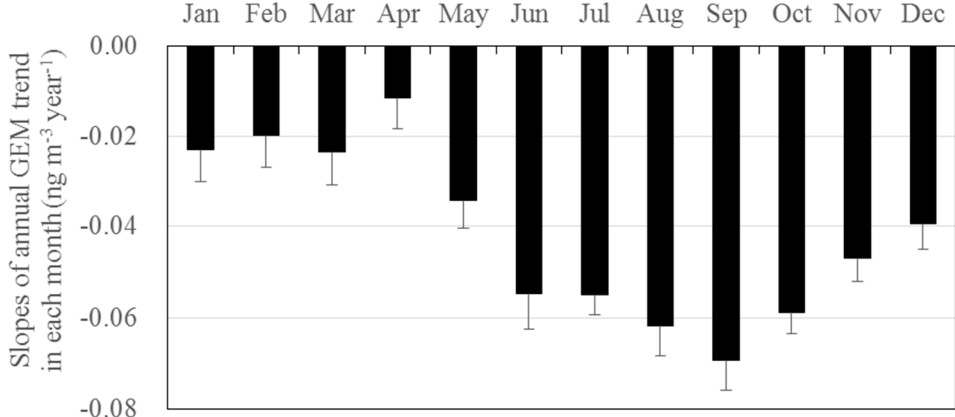

**Figure 12.** Annual trends in the slope of the least-squares regression based on the daily mean of GEM concentrations in each month.

The annual trends of GOM and PBM$_{2.5}$ could not be calculated because 70% of the GOM values and 30% of the PBM$_{2.5}$ values were below their MDLs (see Section 2.2). In addition, a negative bias on Tekran measurements has been reported for Hg loads below 10 pg [61–63]. The bias occurred

due to the instrument's default integration of Hg peaks at Hg loads, which was exacerbated with deceasing Hg loads [62]. As shown in Figure 2b, even if the Hg loads were below 10 pg, our two datasets obtained by the Tekran instrument and the offline manual method using the different CVAFS and CVAAS detectors were in good agreement. Thus, it was expected that the bias probably did not affect the precision of our GOM and $PBM_{2.5}$ measurements. However, their concentrations at Cape Hedo were very low. Furthermore, as mentioned above, the GOM concentrations varied due to the photochemical reactions between GEM and oxidants in the marine boundary layer in the summer and in anthropogenic plumes transported from the Asian continent during the cold season. The $PBM_{2.5}$ concentrations were also elevated majorly as per the regional transport from the Asian continent. Therefore, the seasonal and diurnal variations in the GOM and $PBM_{2.5}$ concentrations were larger than the variation in GEM concentrations. In addition, it is well known that their atmospheric residence times are short as compared to that for GEM. Therefore, it seems to be more difficult to detect their long-term trends compared to GEM.

## 4. Conclusions

Atmospheric GEM is recognized as a major global pollutant because of its capability to be transported over long distances and to be deposited into marine and terrestrial surfaces far from its emission sources after the conversions to GOM and PBM. The monitoring of atmospheric Hg concentrations started at several sites in Western Europe and North America in the 1980s; since then, the activity has expanded all over the world. In recent years, some researchers reported the long-term trends of GEM or TGM concentrations that spanned over several decades. Decreasing trends were observed in Europe and North America, whereas no significant or weak decreasing trends were observed in the Arctic regions. In this study, we continuously measured the concentrations of atmospheric GEM, GOM, and $PBM_{2.5}$ for more than 10 years at Cape Hedo in the Okinawa islands that are located at the marginal sea area adjacent to the eastern part of the Asian continent, where anthropogenic Hg emissions are the largest. The overall downward trend in GEM concentrations at Cape Hedo was comparable to the trends observed at other sites in Europe and North America. In addition, the rate of decrease was weaker during the cold season than during the warm season. Meanwhile, we could not find the long-term trends on GOM and $PBM_{2.5}$ concentrations. During the cold season, the GEM, GOM, and $PBM_{2.5}$ concentrations increased due to the regional transport from the Asian continent. Therefore, these findings suggest that such decline of the GEM concentrations at Cape Hedo was caused by the influence of a decreasing trend at the global-scale. Furthermore, this influence was restrained during the cold season by the regional transport from the Asian continent. However, further investigations with long-term observations of oceanic Hg in the East China Sea and the North Pacific Ocean and long-term trend analyses using global models are needed in order to delineate the atmospheric Hg trends and the interaction between atmospheric and oceanic Hg. Since the Minamata Convention came into force in 2017, each country is responsible for the reduction of Hg use and emissions. As Asia is one of the main contributors to global Hg emissions, monitoring of atmospheric and oceanic Hg in this part of the world is of particular concern. Therefore, Hg monitoring activities have to be continued in the next decades to understand fate and cycle of Hg at regional and global scales and to evaluate the effectiveness of the Minamata Convention.

**Author Contributions:** K.M. analyzed and interpreted the data and wrote the paper. N.S., Y.S., A.T., A.T., N.F., K.K., and A.M. interpreted the data, provided advice for data analysis, and revised the paper. S.K. provided the data on CO and $O_3$ and provided advice for data analysis. T.Y. and J.C. performed data reduction and analyses. T.H. and H.N. carried out the operation and maintenance of the speciated Hg monitoring system. M.S. defined the research theme and interpreted the data. All authors approved the final draft paper and agreed to be accountable for all aspects of this work.

**Funding:** This research received no external funding.

**Acknowledgments:** The atmospheric speciated mercury data used in this study were acquired from the "Background Monitoring Survey for Atmospheric Mercury and Other Metal Element Concentrations in Aerosols"

conducted by the Ministry of the Environment, Japan. We thank Aya Iwasaki in Okinawa Prefecture Institute of Health and Environment for helping with CO and O$_3$ measurements.

**Conflicts of Interest:** The authors declare no conflicts of interest.

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
