# Peer review of "Long-Term Observation of Atmospheric Speciated Mercury during 2007–2018 at Cape Hedo, Okinawa, Japan"

_atmosphere, doi:10.3390/atmos10070362_

Round 1

Reviewer 1 Report

I thought you did an excellent job on the paper. I just have a few notes (below). The major changes I am going to suggest are editorial in nature, having to do with the use of English. And these are only minor change suggestion for readability. Please feel free to use them if you think best. I have included them as track changes in a word document and attached this document. Overall the document is in very very good shape.

Comments.

1.       Methods section; I am more familiar with the methods section after the introduction and before the results. I don’t know what the journal’s style is. That will be the editors call of where the methods section goes. Additionally, I would add the basic description of the methods for trends analysis here. A couple of sentences should be fine.

2.       Page 8, line 8. I assume you mean monthly median values here? There are a large number of observations for your trend line; many more than annual median values.

3.       Page 8, line 10; significant figures in the trends rate; I would suggest two are plenty.

4.       Figure 8 x axis; these are hard to locate. I think you can use just year as in Figure 9, to make it easier for the reader, and perhaps just put in minor hash marks.

5.       Just a comment on the trends section, Page 10. Given your substitution of the values below detection with ½ of the MDL, this likely biased the trends determination. I would suggest you mention something like this in the discussion. This is particularly true for GOM with 70% less than MDL. It might be that the trends determination is not relevant, or useful with 70% of the values below detection. Same issue with PBM, but not as severe.

6.       Figures: I think it is traditionally a small “p” rather than capital P in trends determination but it may be used both ways.

Reviewer 2 Report

The paper present a unique long-term monitoring of GEM, GOM, and PBM at a remote site in western Pacific Ocean. The data are analysed in terms of trends, seasonal and diurnal variation, and the seasonal variation is explained by photochemistry and different transport patterns.

The manuscript is well written (a more precise wording would be possible at times) and organised, although the section on “Materials and Methods” would be better placed before “Results and Discussion”. The data are unique and deserve to be published. I recommend the publication of the paper with changes proposed below:

The paper reads as if GOM were solely of photochemical origin in summer. This is inconsistent with the figures 3 and 7 which clearly show GOM in polluted air transported from the continent in winter. Correlation of GOM with O3 in during the pollution events shown in figure 7 suggest aged air masses with photochemical formation of O3 (which has no emissions) and possibly also of GOM. But this photochemical GOM formation is due to photochemistry in anthropogenic plumes, not over the ocean. Anthropogenic GOM emission cannot be excluded as there is lot of literature on direct GOM emission e.g. from power plants.  

Section 3 on materials and methods follows usually the introduction. The reason is that in order to understand the results one needs to know how the measurements were made.

Page 1, line 35: Because of the generally higher concentrations in the Northern Hemisphere as described by Sprovieri et al. (2016) measurements there should not be compared with those in Southern Hemisphere.

Page 1, line 39: “Midlatitude regions of the Northern Hemisphere” is perhaps too general. Long-term measurements of 10 and more years exist only in Europe and North America

Page 2, line 11: “Mercury is … deposited to regions far from the emission sources…” – applies only for elemental mercury.

Page 2, lines 12/13: Bioaccumulation occurs only in aquatic nutrition chain.

Page 2, lines 16/17: Solubility is only one factor, slow oxidation and a possible GOM reduction are another ones.

Page 2, lines 20/21: GOM is also emitted – e.g. by power plants, depending on the flue cleaning systems.

Page 2, lines 45/46: Decrease of anthropogenic emissions plays also a role – please note Zhang et al., PNAS, 113, 526-531, 2016.

Page 3, lines 33/34: Bad English: concentrations did not change. The concentrations are elevated because of regional pollution. If the pollution were global then you should measure the same as at Mace Head and other sites in the Northern Hemisphere.

Page 4, line 30: O3 is not a radical.

Page 4, line 40: “decreased” instead of “lowered”

Page 9, lines 3-7: See above (page 2, lines 45/46).

Figure 5 and 6: How would a similar figure for GOM look like? Does it support your hypothesis of predominantly oceanic origin of GOM?

Page 13, lines 16/17: see page 1, line 35

Page 13, lines 20/21: Here I would mention that the overall downward trend at Hedo was comparable to the trends observed at other sites in mid-latitude of Northern Hemisphere.

Section 3: Negative bias of Tekran measurements at loads below 10 pg has been reported (Swartzendruber et al., Atmos. Environ., 43, 3648-3651, 2009; Slemr et al., Atmos. Meas. Tech., 9, 2291-2302, 2016; Ambrose, Atmos. Meas. Tech., 10, 5063-5073, 2017). This may affect especially the GOM and PBM measurements where your average loads amount to 2.04 and 3.12 pg, respectively. Please mention and discuss it in the text. Manual and automatic measurements will correlate if both are measured by Tekran and the loads are similar. Figure 12 b would then provide no proof of the absence of the bias. The bias of the Tekran system affects also the precision of the GOM and PBM measurements and by that it makes more difficult to detect their trends.
